# CONTRASTIVE LEARNING RECOVERS CAUSAL FEATURES FOR INSTRUMENTAL VARIABLE REGRESSION

## ABSTRACT

Instrumental Variable (IV) regression is an established technique for estimating causal effects in the presence of unobserved confounders. A core IV assumption is that we have access to an external variable—called the instrument—which directly influences the treatment variable. In this work, we consider a more challenging yet realistic setting where the treatment is high-dimensional but admits a latent structure, through which it interacts with the outcome. To overcome this problem, we leverage insights from the Independently Modulated Component Analysis (IMCA), which is a framework that relaxes the independence assumption in Independent Component Analysis (ICA). Specifically, we propose a general contrastive learning framework to recover the latent features up to an affine transformation which may be related to the instrument by a (non-)linear function. We prove that the recovered representation is compatible with standard IV techniques. Empirically, we demonstrate the effectiveness of our method using control function and two-stage least squares (2SLS) estimators and evaluate the robustness of the learned estimators in distribution shift setting.

## 1 INTRODUCTION

Conventional supervised learning techniques, such as empirical risk minimization (ERM), are widely used to model relationships between features and outcomes. To correctly capture causal effects of the predictors, these methods rely on the assumption that the residuals of the target variable are independent of the features. This assumption, however, does not generally hold. Consider a setting where we observe a treatment $X$ and an outcome $Y$ which can be expressed as $Y = f_0(X) + \varepsilon$, with $\mathbb{E}[\varepsilon] = 0$ but $\mathbb{E}[\varepsilon|X] \neq 0$. Such a data generative mechanism violates the standard assumption that the noise is independent of the features, leading to $\mathbb{E}[Y|X] \neq f_0(X)$. Thus, classical supervised learning methods fail to recover the true causal effect. To address this, *Instrumental Variable* (IV) regression (Imbens & Angrist, 1994) assumes the observation of an *instrument* that affects the outcome only through the treatment variable and is thus independent from the residuals. While originally formulated for linear functions $f_0$, nonparametric approaches to IV regression (Newey & Powell, 2003; Ai & Chen, 2003; Darolles et al., 2011) have emerged. Nonparametric instrumental variable (NPIV) regression is often categorized into two larger areas which consist of conditional moments methods (Bennett et al., 2019; Saengkyongam et al., 2022; Zhang et al., 2023; Bennett et al., 2023) that aim to solve a min-max optimization problem exploiting the independence of instrument and residuals, and two-stage estimators (Newey & Powell, 2003; Hartford et al., 2017; Chen & Christensen, 2018; Singh et al., 2019; Meunier et al., 2024) that first estimate the relation between instrument $A$ and treatment $X$ and then regress the outcome $Y$ based on the estimation result of stage one. The latter approach has its roots in two-stage least squares (2SLS) (Angrist & Imbens, 1995) discussed in Section 2.

We focus on a setting where the treatment $X$ is high-dimensional but driven by a lower-dimensional latent structure $Z$ through an injective mixing function $g_0 \colon \mathcal{Z} \to \mathcal{X}$, *i.e* $X := g_0(Z)$. The outcome variable $Y$ depends on these latent factors rather than on the raw high-dimensional observations ($X$ influences outcome only through latents, cf. Figure 1a). A practical example is a post-surgery CT scan, where the observed image $X$ reflects underlying factors $Z$ (e.g., surgical precision or tissue response), and the outcome corresponds to the patient's recovery. Prior works in causal inference have also considered high-dimensional structured treatments, including text, images (Kaddour et al.,

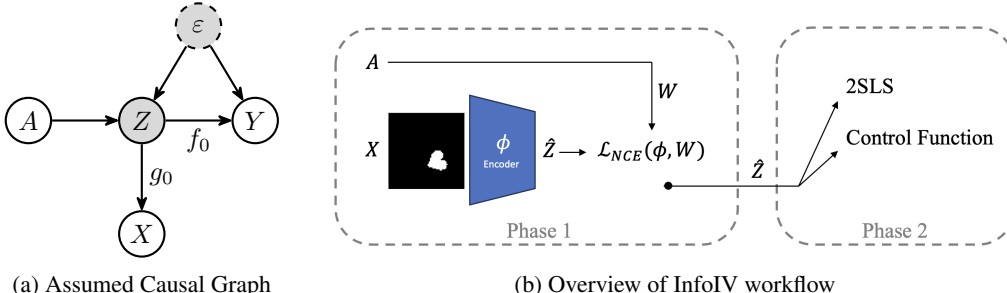

(a) Assumed Causal Graph        (b) Overview of InfoIV workflow

Figure 1: On the left (a) we show the causal graph corresponding to our assumed data generative mechanism, where $Z$ is the latent features, $X$ the observed potentially high-dimensional treatment, $A$ the observed instrument variable, $Y$ the outcome, and $\varepsilon$ represents the unobserved confounder (sometimes implicit in $\varepsilon$). The right plot (b) provides an overview of our method *InfoIV* that learns an encoder $\psi$ inverting $g_0$ in Phase 1 (example picture from dSprites (Matthey et al., 2017)) and supplements the estimated $\hat{Z}$ to either 2SLS or a control function approach in Phase 2.

2021) or even graphs (Harada & Kashima, 2021). Most closely related to our setting are deep feature IV (DFIV) (Xu et al., 2021) and REP4EX (Saengkyongam et al., 2024). DFIV follows the 2SLS approach whereas the regression steps are performed through deep neural networks which are jointly optimized. REP4EX tackles a similar setting as shown in the graph in Figure 1a with the requirement that the function from $A$ to $Z$ is linear. Under these assumptions, REP4EX learns a representation of $Z$ based on an autoencoder and adapts a control variable approach (Newey et al., 1999) to perform intervention extrapolation—connecting to a large body of work that studies causal approaches for out-of-distribution prediction (Rojas-Carulla et al., 2018; Arjovsky et al., 2019; Dominik Rothenhäusler et al., 2021; Shen & Meinshausen, 2024). In this paper, our goal is to both be able to handle nonlinear relations between instrument $A$ and latent variable $Z$, as well as to allow for potentially high-dimensional treatments $X$ that are the result of a nonlinear mixing of the latent features $Z$. To address this challenge, we connect instrument variables with causal representation learning.

Similarly to IV regression, the field of causal representation learning (CRL) (Schölkopf et al., 2021) often relies on some extra information such as an observed auxiliary variable to learn representations that are suitable for performing causal downstream tasks. A core problem in CRL is nonlinear independent component analysis (ICA) (Hyvärinen et al., 2001; 2024), which aims to recover independent sources $Z$ from nonlinearly mixed signals $X$. A central question in ICA is that of identifiability—whether the sources can be recovered from observational data alone. This task is not feasible without additional assumptions on the data generative process (Hyvärinen & Pajunen, 1999). For contrastive learning, (blockwise) identification results have been derived by leveraging self-supervision (Zimmermann et al., 2021; Von Kügelgen et al., 2021), multi-modality (Daunhawer et al., 2023), or multi-view data (Gresele et al., 2020; Yao et al., 2024; Heurtebise et al., 2025). More closely related to our problem are frameworks relying on auxiliary variables. Just as an instrumental variable enables identification of a causal effect, identifiability of nonlinear ICA can be achieved by introducing an auxiliary variable (Khemakhem et al., 2020a, iVAE), under the assumption that the latent variables are independent conditioned on the auxiliary. Potential examples for such an auxiliary variable include, e.g., the time index or the history in temporal data, as well as the class label in a classification context (Hyvärinen et al., 2019). Furthermore, extensions of the iVAE-framework show promising results when adapted to the potential outcomes framework (Wu & Fukumizu, 2022) and the front-door adjustment (Xu et al., 2024).

The *key idea* of our approach is simple yet powerful. We show that under weak assumptions, instruments $A$ can be equivalently used as auxiliary variable to recover the latent features $Z$ up to a linear transformation from $X$ via nonlinear ICA (see Figure 1b). Despite this indeterminacy, the recovered latents can then be plugged into standard approaches based on 2SLS, as well as control functions to estimate the causal effect of $X$ on $Y$, providing a general framework for a range of NPIV approaches. We further show that suitable latent representations can be learned by adopting contrastive learning—in particular the popular InfoNCE objective (van den Oord et al., 2019). Another twist to our approach is that we do not require the strong independence assumption of ICA,

which would restrict the types of confounding that we can account for. Instead, we opt to ground our work in Independent Modulated Component Analysis (IMCA) (Khemakhem et al., 2020b), which provides us with weaker assumptions on the data generative process. To showcase the capabilities of this approach, we introduce our method, called *InfoIV*, and benchmark it in terms of representation learning capabilities both for tabular and image data. Further, we instantiate it via 2SLS and control functions for causal effect estimation and extrapolation, respectively.[1]

The remainder of the paper is organized as follows. In Section 2, we review IV regression. Section 3 links IV regression to representation learning. Subsequently, in Section 4, we propose InfoIV, show its suitability for IV regression and discuss how to instantiate it for 2SLS and control function approaches. In Section 5, we empirically evaluate InfoIV, and we conclude in Section 6.

## 2 INSTRUMENT VARIABLE REGRESSION

Instrument variable (IV) regression assumes that we observe a treatment $X \in \mathcal{X} \subset \mathbb{R}^{d_X}$ and an outcome $Y \in \mathcal{Y}$ generated according to the following structural causal model (SCM)

$$Y := f_0(X) + \varepsilon, \tag{1}$$

where $f_0$ denotes the structural function and $\varepsilon$ is a residual term with zero mean and finite variance. In contrast to the standard supervised learning setting—where $\varepsilon$ are assumed to be *i.i.d.* and independent of $X$—the IV framework allows for the presence of confounder, which implies that the residual term is correlated with the treatment, *i.e.*, $\mathbb{E}[\varepsilon|X] \neq 0$. In this case, regressing $Y$ on $X$ does not generally identify the true structural function, since $f_0(x) \neq \mathbb{E}[Y|X = x]$. To account for the confounding variable, we assume that we observe an *instrument variable* $A \in \mathbb{R}^{d_A}$ which satisfies the following conditions.

**Assumption 2.1.** An instrument $A \in \mathbb{R}^{d_A}$ satisfies the following conditions: (i) (**Relevance**), *i.e.*, $P(X|A)$ is not constant in $A$. (ii) (**Exogeneity**), *i.e.*, $\mathbb{E}[\varepsilon|A] = 0$.

Based on Assumption 2.1 the ground-truth structural function satisfies $\mathbb{E}[Y|A] = \mathbb{E}[f_0(X)|A]$, which allows us to derive the following prominent result, which we recite for completeness.

**Theorem 2.2** (Newey & Powell (2003)). *Assume $X, Y$ generated according to Equation (1), and let $A$ be an instrument satisfying Assumption 2.1. Further assume that the distribution of $X$ conditional on $A$ is exponential. Then, if $f_0$ and $\hat{f}$ are differentiable, $\mathbb{E}[f_0(X)|A] = \mathbb{E}[\hat{f}(X)|A]$ implies $f_0 = \hat{f}$.*

Simply put, if an estimator $\hat{f}$ reproduces the ground-truth conditional expectation of the structural function given $A$, then it coincides with $f_0$. Since directly minimizing this conditional expectation is generally ill-posed (Nashed & Wahba, 1974), more practical estimators have been derived.

**Two-stage Least Square Estimator.**     To solve for this problem, Newey & Powell (2003) propose to use a two-stage least square (2SLS) regression (Angrist & Imbens, 1995) to optimize the following optimization problem:

$$\hat{f} = \arg\min_{f \in \mathcal{F}} \mathcal{L}(f), \quad \mathcal{L}(f) = \mathbb{E}_{Y,A}[(Y - \mathbb{E}[f(X)|A])^2]. \tag{2}$$

A common approach is to parametrize the structural function as $f_0(x) = \theta^T f(x)$ where $\theta$ is a learnable coefficient vector and $f(x)$ is a dictionary of functions (Newey & Powell, 2003; Blundell et al., 2007; Chen & Christensen, 2018). In the first stage, 2SLS estimates $\mathbb{E}[f(X)|A]$ by regressing $f(X)$ on $A$, and in the second stage the coefficient vector $\theta$ is obtained from the closed-form solution of the linear regression of $Y$ on the estimated $\mathbb{E}[f(X)|A]$. In a linear 2SLS setting the chosen dictionary is the identity $f(x) = x$ while more flexible methods like Kernel IV (Singh et al., 2019) leverage non-linear functions in reproducing kernel Hilbert spaces (RKHS). Those methods, however, suffer from limited expressivity since the dictionary is pre-defined. To address this limitation, DeepIV (Hartford et al., 2017) proposes to leverage neural networks in both stages: first to approximate the conditional distribution of $X$ given $A$, and second to approximate the structural function. Bennett et al. (2019) have shown that those methods usually fail in a high-dimension setting, for example when $X$ is an image. Another approach, deep feature IV (DFIV) overcomes some of this limitations by jointly

---

[1]The code is attached to the submission and will be made publicly available upon acceptance.

optimizing both networks (Xu et al., 2021), yielding an advantage compared to fixed-feature estimators (Kim et al., 2025). To avoid the problem of having to learn a powerful conditional generative model, we instead propose to approximate the conditional distribution in the latent space.

**Control Function Estimator.** While 2SLS ignores the residual variation, the control function approach explicitly models the endogenous noise associated with the treatment and uses it as an additional regressor in the outcome model. For intuition, consider the SCM in Equation (1) and assume we observe an instrument $A$ that satisfies the conditions in Assumption 2.1. Further, suppose that the treatment and outcome are confounded through a residual term, *i.e.*, $X := h(A) + V$, where the residual term of $Y$ is $\varepsilon := l(V) + \eta$. Thus, the conditional expectation of $Y$ given $X$ and $V$ yields:

$$\mathbb{E}[Y \mid X, V] = \mathbb{E}[f_0(X) + l(V) + \eta \mid X, V] = f_0(X) + l(V), \tag{3}$$

since $\mathbb{E}[\eta \mid Z, V] = 0$. This equality motivates the *control function* method. First, we regress $X$ on $A$ to obtain the predicted component $X_A = \mathbb{E}[X \mid A]$, analogous to the first stage of 2SLS. Then, because $A \perp\!\!\!\perp V$, the residuals can be consistently estimated as $\hat{V} := X - X_A$. Finally, we perform an additive regression of $Y$ on $X$ and $\hat{V}$ to estimate $f_0$ and $l$. In particular, Newey et al. (1999) showed that, under the assumption that $f_0$ and $l$ are differentiable, the ground-truth causal effect $f_0$ can be recovered up to an additive constant. Further, Saengkyongam et al. (2024) show that the control function approach could be leveraged in order to perform extrapolation over unseen values of $A$, under the assumption that treatment $X$ and instrument $A$ are linearly related. In comparison to 2SLS, however, the instrument has to be available at test time.

## 3 DATA GENERATIVE PROCESS

In contrast with the classic IV setting introduced previously, we consider a *representation-based* variant where the treatment $X \in \mathcal{X}$ is high-dimensional and admits a lower-dimension latent representation $Z \in \mathcal{Z} \subset \mathbb{R}^{d_Z}$. The outcome $Y \in \mathcal{Y} \subset \mathbb{R}$ only depends on treatment $X$ through latent factors $Z$ and an *instrument variable* $A \in \mathcal{A} \subset \mathbb{R}^{d_A}$ is observed. A summary of the corresponding causal graph is provided in Figure 1a. Throughout the paper, we assume that our data are generated according to the following SCM:

$$\mathcal{S} : \begin{cases} X := g_0(Z) \\ Y := f_0(Z) + \varepsilon, \end{cases} \tag{4}$$

where $\varepsilon$ is a residual term with zero mean and finite variance but correlated with latent factors $Z$, *i.e.*, $\mathbb{E}[\varepsilon|Z] \neq 0$, $g_0 : \mathcal{Z} \to \mathcal{X}$ is a nonlinear injective mixing function, and $f_0 : \mathcal{Z} \to \mathbb{R}$ is the structural function. Since $Z$ is not observed, our first goal is to recover the latent features $Z$ up to some indeterminacy exploiting the instrument $A$ as an auxiliary variable.

### 3.1 INSTRUMENT- AND AUXILIARY VARIABLES

It is well-known that in the general case, nonlinear ICA is infeasible (Hyvärinen & Pajunen, 1999), however, the instrument variable setting as introduced before assumes the observation of a variable $A$ with direct causal influence on $Z$. Similarly, the nonlinear ICA literature often relies on an observed *auxiliary variable* Hyvärinen et al. (2019) with direct causal influence on latent variable to guarantee its identifiability. We build upon theory from Khemakhem et al. (2020a;b) to show that the latent features $Z$ can be recovered up to an affine and pointwise transformation, with a few assumptions on the distribution of $Z$ which are compatible with the general IV framework. Let us define an encoder $\phi : \mathcal{X} \to \mathcal{Z}$, typically parametrized as a neural network, whose goal is to approximate the inverse mixing function $g_0^{-1}$. We define *affine identifiability* as (introduced in Saengkyongam et al. (2024)):

**Definition 3.1** (Latent Identifiability). We say that the latent features $Z$ are identified up to an affine transformation and pointwise transformation if there exist an encoder $\phi : \mathcal{X} \to \mathcal{Z}$ such that:

$$\phi \circ g_0(z) = RT(z) + c, \forall z \in \mathcal{Z}$$

with $T$ a pointwise function, $R$ an invertible matrix and $c \in \mathbb{R}^{d_Z}$.

While classic identifiability results usually rely on the mutual independence of the $Z$ components when conditioned on $A$, which would restrict the types of confounding that we can consider, we build upon the results of Khemakhem et al. (2020b), who proof identifiability in a more general exponential factorial case. Let us first define the conditional exponential family.

**Definition 3.2** (Conditional Exponentially Factorial Distribution). We say that a multivariate random variable $Z$ is conditional exponentially factorial if its conditional density has the form

$$p_{T,\lambda}(z|a) := \mu(z) \exp\left( \sum_{i=1}^{d_Z} T_i(z_i)^\top \lambda_i(a) - \Gamma(a) \right), \tag{5}$$

where $T_i : \mathbb{R} \to \mathbb{R}^k$ are called the *sufficient statistics*.

*Remark* 3.3. Note that the base measure $\mu(z)$ captures the part of the variation in $Z$ not explained by $A$, *i.e.*, the confounding. Due to this component, we do not have to assume that the components in $Z$ are conditionally independent given $A$. Further, the distributional assumption is rather general, as the exponential family includes a lot of classic distributions like Gaussian, Binomial, Beta and Chi-deux.

Next, we show that under these model assumptions, we can extend the identification result of Khemakhem et al. (2020b) to our instrument variable setting, and show that InfoNCE (van den Oord et al., 2019) is a suitable loss to train an encoder satisfying Definition 3.1.

## 4 INFOIV

For the data generative process defined in the previous section, we propose a two-phase method to perform IV regression and extrapolation which we sketch in Algorithm 1. In Phase 1, the instrument $A$ is used as an auxiliary variable to recover the sufficient statistic of the latent variable $Z$ up to an invertible affine transformation (cf. Section 4.1). Specifically, we train an encoder $\phi$ by minimizing a contrastive loss inspired from InfoNCE (van den Oord et al., 2019), for which we prove that it identifies the true inverse mixing function $g_0^{-1}$ up to an affine transformation and coordinatewise nonlinearities defined by the sufficient statistics. Subsequently, in Section 4.2, we show that we can leverage the learned representations for a 2SLS approach (Phase 2a), as

---

**Algorithm 1:** InfoIV (Sketch)

**input :** Data drawn from $P(A, X, Y)$
// **Phase 1 (Representation Learning)**
1. Obtain $\phi^*, W^* = \arg\min_{\phi, W} \mathcal{L}_{\text{NCE}}(\phi, \psi)$
2. Estimate latent variable $\hat{Z} = \phi^*(X)$
// **Phase 2a (2SLS)**
3. Estimate $\mathbb{E}[\hat{Z}|A]$— obtaining $\hat{Z}_A$
4. Estimate $\hat{f}_0$ from the regression of $Y$ on $\hat{Z}_A$
// **Phase 2b (Control Function)**
5. Estimate $\mathbb{E}[\hat{Z}|A]$— obtaining $\hat{Z}_A$
6. Obtain $\hat{V} = \hat{Z} - \hat{Z}_A$
7. Estimate $\hat{f}_0, \hat{l}$ from the additive regression of $Y$ based on $\hat{Z}$ and $\hat{V}$

---

well as for extrapolation (Section 4.3) via the control function approach (Phase 2b) similar to the autoencoder-based method proposed by Saengkyongam et al. (2024). The overall workflow of InfoIV is also sketched in Figure 1b.

### 4.1 RECOVERING SUITABLE REPRESENTATIONS FOR IV REGRESSION

To recover $Z$ up to a permutation suitable for IV regression, we train an encoder $\phi$ to maximize the similarity between our estimated latent representations $\hat{z} := \phi(x)$ and its corresponding instrument $a$. Accordingly, we modify the well-known InfoNCE loss as follows:

$$\mathcal{L}_{\text{NCE}}(\phi, W) = \mathbb{E}_{A,X} \left[ -\log \frac{e^{-\phi(X)WA/\tau}}{\sum_{\tilde{A} \sim P_A} e^{-\phi(X)W\tilde{A}/\tau}} \right], \tag{6}$$

where $W$ is a learnable matrix $\in \mathbb{R}^{d_Z \times d_A}$ and $\tau$ is the temperature.

We show that under assumptions of sufficient variability of $Z$ *w.r.t.* the auxiliary variable $A$, upon convergence of the loss, the corresponding encoder weakly identifies the latent features $Z$.

**Theorem 4.1.** *Let the conditional $Z \mid A$ follow the conditional factorial distribution introduced in Definition 3.2, with parameters $(T, \lambda)$. Further, let $g_0 : \mathcal{Z} \to \mathcal{X}$ be a (non-linear) injective mixing function and $X := g_0(Z)$. Consider that the following conditions hold:*

1. *The sufficient statistic $T(z) = (T_i(z_i))_{i=1}^{d_Z}$ is differentiable almost everywhere.*
2. *There exist $d_Z + 1$ distinct points $u^0, ..., u^{d_Z}$ such that the matrix*

$$L_\lambda(\mathbf{u}) = (\lambda(u^1) - \lambda(u^0), ...., \lambda(u^n) - \lambda(u^0)) \quad \text{is invertible.}$$

3. *We train $\phi^*$ an encoder with universal approximation capability and $W^* \in \mathbb{R}^{d_Z \times d_A}$ on the loss stated in Equation (6).*

*Then in the limit of infinite data, $\phi^*(X)$ identifies $Z$ up to an invertible linear transformation and pointwise nonlinearities defined by its sufficient statistics.*

*Remark* 4.2. Hyvärinen et al. (2019) introduce a related contrastive loss that enables weak identification of latent variables under the same assumptions. Their method trains a logistic regression head on top of the encoder, using as input both the learned latent representation and the instrument, in order to discriminate between positive pairs (sampled from the joint distribution) and negative pairs (sampled independently). The weak identifiability of this approach in the general conditional exponentially factorial distribution was established by Khemakhem et al. (2020b). In contrast, we show experimentally that our method based on the InfoNCE loss converges faster. We hypothesize that this improvement arises because, at each SGD iteration, our approach compares each point against all other negative pairs within the batch, making it computationally more stable.

### 4.2    INFOIV-2SLS

The previous result establishes that we can recover the latent features up to an invertible linear transformation of the sufficient statistic in the conditional exponential case. We now show that this level of indeterminacy suffices to uniquely identify the causal effect, by extending Theorem 2.2 (Newey & Powell, 2003).

**Lemma 4.3.** *Let $(Z, Y)$ be generated according to Equation (4). Suppose we observe an instrument $A$ that satisfies Assumption 2.1 with respect to $Z$. Let $T$ be differentiable almost everywhere, $R$ an invertible matrix, and $c$ a vector, defining a mapping $\tau : \mathbb{R}^{d_Z} \to \mathbb{R}^{d_Z}$ by $\tau(z) = R T(z) + c$. Then:*

$$\mathbb{E}[f_0(Z) \mid A] = \mathbb{E}[\hat{f} \circ \tau(Z) \mid A] \quad \Rightarrow \quad f_0 = \hat{f} \circ \tau. \tag{7}$$

*Proof.* Since $R$ is invertible and $T$ is differentiable almost everywhere, the mapping $\tau$ is differentiable almost everywhere as well. Hence, both $f_0$ and $\hat{f} \circ \tau$ are differentiable and satisfy the conditions of Newey & Powell (2003). By the completeness property of the exponential family, the conditional expectation equality implies the functional equality, establishing the claim. $\square$

In other words, if the estimator $\hat{f}$ applied to the approximated latent representation $\tau(z)$ correctly captures the conditional distribution, then the composed mapping approximates the true structural function. In this case, the latent indeterminacy cancels out, enabling recovery of the true causal effect from $X$ to $Y$:

$$\hat{f} \circ \phi = \hat{f} \circ \tau \circ g_0^{-1} = f_0 \circ g_0^{-1}, \tag{8}$$

by noting that our encoder recovers the ground-truth features up to $\tau$ and thus $\phi = \tau \circ g_0^{-1}$.

In summary, Theorem 4.1 and Lemma 4.3 ensure that 2SLS approaches are applicable on the learned representation that we recover based on the loss stated in Equation (6). Additionally to standard IV assumptions, $A$ has to fulfill the IMCA assumption with respect to $Z$ (Definition 3.2). As outlined in Algorithm 1, we first train the encoder and subsequently perform 2SLS. In practice, we perform both regression steps independently with neural networks.

### 4.3    INFOIV-CF

We now show that Phase 1 of InfoIV also recovers suitable features for extrapolation tasks, where we aim to predict the result of an intervention on an action variable $A$, when this intervention was not observed in the training support. Using do-notation (Pearl, 2009), this corresponds to estimating

$\mathbb{E}[Y|do(A := a^*)]$. In particular, we build upon the results of Saengkyongam et al. (2024) who relied on an autoencoder trained via moment constraints to obtain the latent features. Saengkyongam et al. (2024) show that one can extrapolate over unseen values of $A$ if we restrict the effect of $A$ on $Z$ to be linear. In particular, let us consider the following SCM:

$$\mathcal{S}_1 : \begin{cases} Z := M_0 A + V \\ X := g_0(Z) \\ Y := f_0(Z) + l(V) + \varepsilon, \end{cases} \tag{9}$$

with $A \perp\!\!\!\perp V, \varepsilon$ whose support's interior is convex. Here, $\varepsilon$ is a noise term with zero mean and finite variance independent from $Z$. We further assume $M_0 \in \mathbb{R}^{d_Z \times d_A}$ to be full-rank and $g_0$ injective. Note that in comparison to Equation (4), the dependence to the confounder $V$ is modeled explicitly, while previously it was absorbed in the noise term.

Most relevant for us is that Saengkyongam et al. (2024) show that if we can train an encoder $\phi$ that *recovers $Z$ up to an affine-transformation*, then one can leverage the control function approach to estimate the true causal-effect $f_0$ and perform extrapolation on $A$, *i.e.*, estimate $\mathbb{E}[Y|do(A := a^*)]$ for all $a^* \in \mathcal{A}$.[2] Consequently, we need to show that we can recover the latent features $Z$ up to an affine-transformation for the SCM above.

**Corollary 4.4.** *Assume $Z := M_0 A + V$ with $M_0$ full-rank and $V \sim \mathcal{N}(0, \Sigma)$. Let $X := g_0(Z)$ with $g_0$ an injective function. Assume that there exist $d_Z + 1$ linearly independent distinct points in $\mathrm{supp}(A)$. Then, in the limit of infinite data an encoder $\phi^*$ trained to minimize loss Equation (6) provides a consistent estimator of $Z$ up to an invertible affine transformation.*

As can be noted, in comparison to 2SLS, we need to restrict the function from $A$ to $Z$ to be linear and need to add some distributional assumptions to ensure that the extrapolation task is well-defined. Similar to 2SLS, all regression steps are performed independently based on neural networks. This concludes our theoretical results. Next, we empirically evaluate the different components of InfoIV.

## 5 EXPERIMENTS

In the following, we compare InfoIV to state-of-the-art approaches for IV regression for tabular and image data, and extrapolation, as well as evaluate InfoIV purely for representation learning. We start with the tabular setting (Section 5.1), then we evaluate our approach on a synthetic image experiment (Section 5.2), and last we evaluate its extrapolation capacity (Section 5.3).

### 5.1 SIMULATING ON CORRELATED CONFOUNDING

For the experiments shown in Figure 2a and Table 1, we simulate data according to the following data-generating process. The instrumental variable $A$ is drawn independently from a uniform distribution. The latent variable $Z$ is then generated according to a conditionally exponential family distribution as defined in Definition 3.2: $Z := \tilde{\mu}(A) + \mathrm{diag}\left(\tilde{\sigma}_1(A), \ldots, \tilde{\sigma}_{d_Z}(A)\right) \varepsilon$, where $\varepsilon \sim \mathcal{N}(0, \Sigma)$ is sampled independently of $A$. The functions $\tilde{\mu}$ and $\tilde{\sigma}_i$ are nonlinear mappings $\mathbb{R}^{d_A} \to \mathbb{R}^{d_Z}$, implemented as randomly initialized neural networks.

Here, $\varepsilon$ corresponds to the base measure $\mu(z)$, *i.e.*, the part of the variation in $Z$ not explained by $A$. In particular, if we enforce conditional independence of the components of $Z$ given $A$, we set $\Sigma$ to be diagonal, so that $\varepsilon$ follows an isotropic Gaussian distribution. Since we consider a more general case, we instead draw $\Sigma$ as a symmetric positive-definite matrix. The observed treatment is then defined as $X := g_0(Z)$, where $g_0$ is a neural network with enforced injectivity. Finally, the outcome variable is generated as $Y := f_0(Z) + \rho R \varepsilon + \eta$, where $R \in \mathbb{R}^{d_Z}$ is a vector, $\eta$ is Gaussian noise, and $\rho \in [0, 1]$ is a parameter controlling the strength of confounding. Additional details about the data-generating process are provided in Section B.2. Prior to all experiments, we evaluate and fix the temperature $\tau$ of the InfoNCE loss as described in Section B.4.

---

[2]For completeness, we recite a shortened version of their theorem in Section A.1.

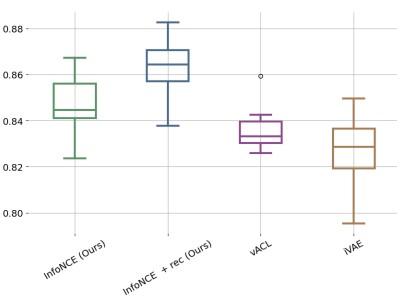 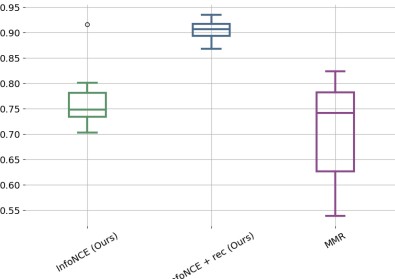

(a) Average MCC (latent recovery).  (b) Average $R^2$ (extrapolation task).

Figure 2: [Latent Recovery] On the left (a), we show the results for latent recovery in terms of MCC (higher is better) for the experiment described in Section 5.1.1. The plot on the right (b) shows the recovery of the latent features in terms of $R^2$ in comparison to MMR (used within REP4EX) for the experiment described in Section 5.3.

### 5.1.1 RECOVERING LATENT FEATURES

We first evaluate Phase 1 of InfoIV, *i.e.*, we evaluate how well we can recover the latent factors by exploiting the instrument $A$ as a proxy variable. As detailed in Section 4.1 this step is performed by minimizing an adaptation of the InfoNCE loss tailored to our setting. We further ablate our method by adding a decoder and a reconstruction term to the loss (cf. Section B.4). Both variants are benchmarked against two baselines: iVAE (Khemakhem et al., 2020a) and vanilla auxiliary contrastive learning (vACL) (Hyvärinen et al., 2019), whose descriptions and implementation details are provided in Section B.1. An advantage of our method is its efficiency: it requires training only a matrix of dimension $d_Z \times d_A$ on top of the encoder, unlike most latent identification methods that require training a decoder (Khemakhem et al., 2020a; Saengkyongam et al., 2024) or a logistic regression head (Hyvärinen et al., 2019).

We sample 20 datasets with $5,000$ data points each, where we set the dimensions $d$ of the involved variables so that $d_Z = 8, d_A = 10$, and $d_X = 12$. Each method is trained for 50 epochs and we report the mean correlation coefficient (MCC) of the estimated latent variables with the ground-truth $Z$ after aligning them with the optimal affine transformation. A higher MCC indicates better recovery of the true latent structure. The results are shown in Figure 2a. We see that our InfoNCE variant to perform Phase 1 of Algorithm 1 outperforms both iVAE and vACL. Our ablation study in which we add a decoder and a reconstruction term to the loss, provides additional benefits, increasing the mean MCC by approximately 0.015. While helping in terms of reconstruction, however, we observe that increasing the weight for the reconstruction term decreases the performance for the estimation of causal effects, as shown in Section B.4.

### 5.1.2 RECOVERING CAUSAL EFFECT

Once the latent variable is recovered up to an acceptable indeterminacy, we proceed to estimate the causal effect. In particular, we always train for 50 epochs in Phase 1. To estimate the causal effect, we proceed in two stages: First, we regress the estimated latent features on the instrument $A$ to obtain a proxy latent. Second, we regress this proxy variable on the outcome $Y$ to recover the causal effect. In both stages, we train neural networks using the standard mean squared error (MSE) loss (cf. Section B.4). To evaluate the ability of our method to recover the true causal effect, we compute the out-of-sample mean squared error (o.o.s. MSE), defined as

$$\text{MSE}_{\text{oos}} = \frac{1}{n} \sum_{i=1}^{n} \|\hat{y}_i - f_0(z_i)\|^2, \tag{10}$$

where $\hat{y}_i$ denotes the models prediction and $f_0(z_i)$ the ground-truth unconfounded outcome corresponding to $f_0(g_0^{-1}(x_i))$. We generate 10 datasets for each confounder strength $\rho \in \{0.1, 0.5, 1\}$ generating $5,000$ datapoints each.

We compare InfoIV-2SLS to the state-of-the-art for nonparametric IV regression, *i.e.* KIV (Singh et al., 2019), DeepGMM (Bennett et al., 2019), and DFIV (Xu et al., 2021) and show the results in

| Method | $\rho = 0.1$ | $\rho = 0.5$ | $\rho = 1$ |
|---|---|---|---|
| 2SLS | $(1.18 \pm 0.2) \times 10^{-2}$ | $(2.00 \pm 0.3) \times 10^{-2}$ | $(2.56 \pm 0.32) \times 10^{-2}$ |
| DeepGMM | $(1.11 \pm 0.08) \times 10^{-3}$ | $(4.44 \pm 0.12) \times 10^{-3}$ | $(5.25 \pm 0.10) \times 10^{-3}$ |
| DFIV | $(\mathbf{4.49 \pm 0.07}) \times \mathbf{10^{-4}}$ | $(\mathbf{1.13 \pm 0.07}) \times \mathbf{10^{-3}}$ | $(1.73 \pm 0.15) \times 10^{-3}$ |
| KIV | $(1.11 \pm 0.06) \times 10^{-3}$ | $(1.17 \pm 0.08) \times 10^{-3}$ | $(\mathbf{1.19 \pm 0.07}) \times \mathbf{10^{-3}}$ |
| InfoIV-2SLS (ours) | $(1.11 \pm 0.12) \times 10^{-3}$ | $(2.24 \pm 0.30) \times 10^{-3}$ | $(3.35 \pm 0.30) \times 10^{-3}$ |

Table 1: MSE$_{\text{oos}}$ results (mean $\pm$ std) across different $\rho$ values (confounder strength). Each method is trained for 100 epochs on 5,000 data points. Bold values indicate best performance per column.

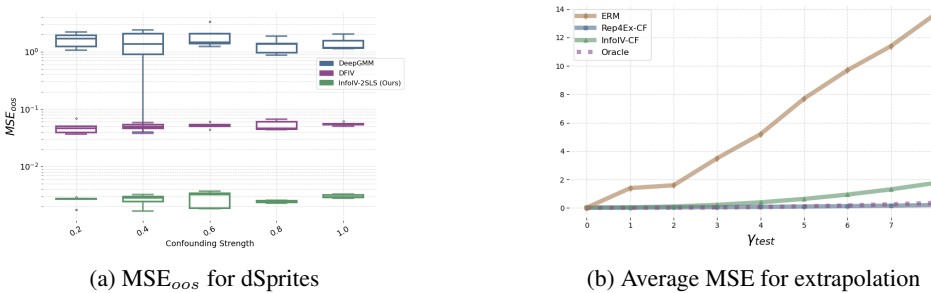

(a) MSE$_{oos}$ for dSprites         (b) Average MSE for extrapolation

Figure 3: [Left] Figure (a) shows the performance for causal effect estimation on the dSprites example in terms of mean MSE$_{oos}$. InfoIV-2SLS clearly outperforms the baselines. [Right] Figure (b) show the result on the extrapolation task with increasing shift $\gamma$. We compare InfoIV-CF to REP4EX, an oracle and a naive baseline (ERM). InfoIV-CF is on par with REP4EX up to $\gamma = 4$.

Table 1. We note that although the second phase of InfoIV-2SLS is not highly optimized in comparison to the baselines, our method still achieves comparable results. Especially when compared to regular 2SLS implemented with neural networks of similar architectures as InfoIV-2SLS directly applied on observed treatment $X$, we see that all methods under comparison achieve an error that is one order of magnitude lower than for this baseline. When moving to image data, in the next section, we showcase the advantages of InfoIV-2SLS.

## 5.2 IV Regression on Image Data

To evaluate our method in the context of high-dimensional treatments, we conduct experiments on the dSprites dataset (Matthey et al., 2017), where each $64 \times 64$ image is described by five latent factors: scale, rotation, shape, x-position, and y-position. In our setup, the treatments $X$ are the images, while the outcome $Y$ is a scalar function of the latent factors $Z$, confounded by the y-position variable (details are provided in Section B.3).

We compare InfoIV-2SLS to DeepGMM Bennett et al. (2019) and DFIV Xu et al. (2021). We adapt the same training procedure for Phase 1 (train for 50 epochs). We used convolution layers for feature extractor, all methods were run with a similar architecture. Each method on 5,000 data points. We trained InfoIV-2SLS and DFIV for 100 epochs and DeepGMM for 50 epochs since it tended to overfit quickly. Results are reported over 10 different random seeds and for different confounding strength. We observe that our method outperforms both DFIV and DeepGMM by orders of magnitude, while DeepIV and KIV failed to converge to reasonable solutions and are therefore excluded from the plot.

## 5.3 Extrapolation

We also evaluate our method in the REP4EX setting (Saengkyongam et al., 2024), where we assume linearity between features and instrument. Particularly, we evaluate the capacity of the *control function* approach to perform extrapolation. We sample data according to the SCM provided in Equation (9), where $g_0$ is an injective neural network, $f_0$ and $l$ are MLPs, $M_0$ is a full-rank matrix and $V$ and $\varepsilon$ are uncorrelated Gaussian noise variables. For the training data, we sample $A \sim \mathcal{U}([-1, 1]^{d_A})$, where $d_A = 10$, $d_Z = 8$, and $d_X = 12$. We follow the control function approach described in Section 4.3 and evaluate the learned causal effect on an extrapolation task where we

sample $A \sim \mathcal{U}([-(\gamma+1), \gamma+1])$ with $\gamma \in \{0, 1, 8\}$. We sample 5 datasets with $10,000$ observations each and apply InfoIV-CF, REP4EX, and ERM with $L^2$ loss as a naive baseline. Both InfoIV-CF and REP4EX are trained for 50 epochs each in all phases. The results are shown in Figure 3b. We see that the representations learned by InfoIV are suitable for extrapolation via the control function approach—strongly outperforming the naive baseline while only being slightly outperformed by the specialized method REP4EX for shifts larger than 4. We also provide some example plots for extrapolation in Section B.5.

We also verify that our InfoNCE loss satisfies the affine identifiability assumption necessary to perform extrapolation (cr. Figure 2b). We compare it to the MMR loss employed by REP4EX, which is outperformed by both of our variants based on InfoNCE.

## 6 DISCUSSION AND CONCLUSION

In this paper, we study a representation-based setting for instrumental variable regression in which the treatment is high-dimensional and admits latent factors that interact with the outcome. Within this setting, we proved the suitability of a two-phase approach in which we first recover the latent features of the treatment up to an affine transformation via a variant of contrastive learning that leverages the instrument as an auxiliary. We implement our method, InfoIV, which exploits the learned latent variables for IV regression via 2SLS, and for extrapolation based on a control function approach in Phase 2 of InfoIV. To recover the latent factors in Phase 1, we adapt the InfoNCE loss to our setting. Through an extensive empirical evaluation, we demonstrate that InfoIV is on par with state-of-the-art 2SLS approaches on tabular data while having an advantage on image data. Further, we demonstrate that InfoIV can be used for extrapolation, being only marginally outperformed by REP4EX specializing on this task.

For future work, we aim to evaluate the extrapolation capacities of InfoIV on vision datasets, as well as explore more principled approaches for 2SLS such as DFIV in Phase 2 of our approach. While our method leverages contrastive learning and uses substantially fewer parameters than deep generative–model approaches such as iVAE (Khemakhem et al., 2020a), our identifiability guarantee is currently limited to the noiseless setting in which $X$ is a deterministic function of the latent variable $Z$. Relaxations of this assumption would be interesting to study in future work.

**Reproducibility Statement.** To ensure reproducibility of our work, we followed common guidelines and ensured to run each experiment with multiple seeds, attached the code as a supplementary file to the submission, and provide details to the experimental setup as well as the hyperparameters of InfoIV and all baselines in Appendix B. All proofs of theoretical claims are provided in Appendix A.

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

# Appendix

## Table of Contents

# A  THEORY

## A.1  IDENTIFIABILITY PROOFS

**Theorem 4.1.** *Let the conditional $Z \mid A$ follow the conditional factorial distribution introduced in Definition 3.2, with parameters $(T, \lambda)$. Further, let $g_0 : \mathcal{Z} \to \mathcal{X}$ be a (non-linear) injective mixing function and $X := g_0(Z)$. Consider that the following conditions hold:*

1. *The sufficient statistic $T(z) = (T_i(z_i))_{i=1}^{d_Z}$ is differentiable almost everywhere.*
2. *There exist $d_Z + 1$ distinct points $u^0, ..., u^{d_Z}$ such that the matrix*
$$L_\lambda(\mathbf{u}) = (\lambda(u^1) - \lambda(u^0), ...., \lambda(u^n) - \lambda(u^0)) \quad \textit{is invertible.}$$
3. *We train $\phi^*$ an encoder with universal approximation capability and $W^* \in \mathbb{R}^{d_Z \times d_A}$ on the loss stated in Equation (6).*

*Then in the limit of infinite data, $\phi^*(X)$ identifies $Z$ up to an invertible linear transformation and pointwise nonlinearities defined by its sufficient statistics.*

*Proof.* As argued in van den Oord et al. (2019), in the limit of infinite data with $\phi$ and $\psi$ having universal approximation capacity, if:
$$\phi^*, W^* = \arg\min_{\phi, W} \mathcal{L}_{\text{NCE}},$$
then
$$e^{\phi^*(x)W^*a} \propto \frac{p(x|a)}{p(x)}.$$

Let us recall that we assume $g_0$ to be injective, therefore it admits a left inverse on its image space contained in $\mathcal{X}$ that we denote $g_0^{-1}$. Under the assumption that $g_0^{-1}$ has full-rank Jacobian, one can apply the change of variable formula with the volume matrix vol $A := \sqrt{\det A^T A}$ (Ben-Israel, 1999).

$$\phi^*(x)W^*a = \log c + \log p(x|a) - \log p(x) \tag{11}$$
$$= \log c + \log p_Z(g_0^{-1}(x)|a) - \log p_Z(g_0^{-1}(x)) \tag{12}$$
$$= \log c + \log p(z|a) - \log p(z) \tag{13}$$

We use the change of variable formula to go from 11 to 12 and notice that the Jacobian volumes cancel themselves. We define $c$ the proportionality constant that is not dependent on $a$ or $x$. At line 13 we simply set $z := g_0^{-1}(x)$. By assumption, $\{Z_i\}_{i=1,...,d_Z}$ given $A$ follow an exponential distribution (Definition 3.2), thus, following the proof of Khemakhem et al. (2020b)[Theorem 9]:

$$\phi^*(x)W^*a = \log p_{T,\lambda}(z|a) - \log p_Z(z) + \log c \tag{14}$$
$$= \log c + T(z)\lambda(a) + \log \mu(z) - \Gamma(a) - p(z), \tag{15}$$

By collecting these equations for every $a_k$, $k \in \{0, ..., d_Z\}$ as defined in assumption 3. and taking out the case $a_0$, we obtain for all $k \in \{1, ..., d_Z\}$:

$$\phi^*(x)W^*(a_k - a_0) = T(z)(\lambda(a_k) - \lambda(a_0)) + (\Gamma(a_0) - \Gamma(a_k)), \tag{16}$$

which yield the following matrix form:

$$\phi^*(x)\Psi = T(z)L + C, \tag{17}$$

with $\Psi$ a $\mathbb{R}^{d_Z \times d_A}$ matrix whose $k$-th row is given by $a_k - a_0$ which is non-zero by assumption, $L$ is defined as in assumption 3 and $C$ is a vector of dimension $d_Z$ whose $k$-th element is given by $\Gamma(a_0) - \Gamma(a_k)$. By assumption, $L$ is invertible thus we can multiply both side by its inverse, which yields the following result:

$$\phi^*(x)R = T(z) + \tilde{C}, \tag{18}$$

with $\tilde{C} := CL^{-1}$ and $R := \Psi L^{-1}$.

Finally, by assumption $T$ has full-rank Jacobian and is thus non-degenerate. As a consequence, the mapping $z \mapsto zR$ has to cover the full-space and thus cannot be degenerate. Since $R$ is a square matrix we deduce its invertibility.

$\square$

After stating the identifiability of InfoNCE in the general IMCA case, we can now state its more refined identifiability in the Gaussian case. Since this result is required for extrapolation, we first recite the corresponding theorem enabling extrapolation of Saengkyongam et al. (2024).

**Theorem A.1** (Saengkyongam et al. (2024), Theorem 4). *Assume Setting 9 with $f_0$ and $l$ differentiable. Let $\phi$ be an encoder that identifies $g_0^{-1}$ up to an affine transformation. Let:*

$$(W_\phi, \alpha_\phi) := \underset{W \in \mathbb{R}^{d_Z \times d_A}, \alpha \in \mathbb{R}^{d_Z}}{\arg\min} \mathbb{E}[\|\phi(X) - (WA + \alpha)\|^2]. \tag{19}$$

*and the estimated noise term $V_\phi := \phi(X) - (W_\phi A + \alpha_\phi)$. Finally, let $\nu$ and $\psi$ be the the estimated functions obtained from additive regression of $Y$ on $\phi(X)$ and $V_\phi$. Then:*

$$\forall a^* \in \mathcal{A}, \mathbb{E}[Y|do(A = a^*)] = \mathbb{E}[\nu(W_\phi a^* + \alpha_\phi + V_\phi)] - (\mathbb{E}[\nu(\phi(X))] - \mathbb{E}[Y]). \tag{20}$$

**Corollary 4.4.** *Assume $Z := M_0 A + V$ with $M_0$ full-rank and $V \sim \mathcal{N}(0, \Sigma)$. Let $X := g_0(Z)$ with $g_0$ an injective function. Assume that there exist $d_Z + 1$ linearly independent distinct points in $supp(A)$. Then, in the limit of infinite data an encoder $\phi^*$ trained to minimize loss Equation (6) provides a consistent estimator of $Z$ up to an invertible affine transformation.*

*Proof.* Let us recall that we sample data from the following SCM:

$$\mathcal{S}: \begin{cases} V \sim \mathcal{N}(0, \Sigma) \\ Z := M_0 A + V \\ X := g_0(Z) \end{cases}$$

with $g_0$ injective and $M_0$ full row rank. We have:

$$p(z|a) = p_V(z - M_0 a) \tag{21}$$

$$= (2\pi)^{-d_Z/2} \det(\Sigma)^{-1/2} \exp\left[-\frac{1}{2}(z - M_0 a)^T \Sigma^{-1}(z - M_0 a)\right] \tag{22}$$

$$= (2\pi)^{-d_Z/2} \det(\Sigma)^{-1/2} \exp\left[-\frac{1}{2}\left[z^T \Sigma^{-1} z - z^T \Sigma^{-1} M_0 a - a^T M_0^T \Sigma^{-1} z + a^T M_0^T \Sigma^{-1} M_0 a\right]\right] \tag{23}$$

$$= (2\pi)^{-d_Z/2} \det(\Sigma)^{-1/2} \exp\left[-\frac{1}{2} z^T \Sigma^{-1} z\right] \exp\left[z \Sigma^{-1} M_0 a\right] \exp\left[-\frac{1}{2} a^T M_0^T \Sigma^{-1} M_0 a\right] \tag{24}$$

$$= \mu(z) \exp\left[z \Sigma^{-1} M_0 a - \Gamma(a)\right] \tag{25}$$

where we go from Eq. 23 to 24 by noticing that the two terms are scalar and the transpose of the other, in Eq. 25 we set $\mu(z) := (2\pi)^{-d_Z/2} \det(\Sigma)^{-1/2} \exp\left[-\frac{1}{2} z^T \Sigma^{-1} z\right]$ and $\Gamma := \frac{1}{2} a^T M_0^T \Sigma^{-1} M_0 a$. This derivation allows us to identify a conditional exponential family with parameters $(T, \lambda)$, as introduced in Definition 3.2. In particular, we obtain $\forall i = 1, ..., d_Z$:

$$\begin{cases} T_i(t) = t, & \forall t \in \mathbb{R} \\ \lambda_i(u) = \Sigma^{-1} M_0 u, & \forall u \in \mathbb{R}^{d_A} \end{cases}$$

It remains to prove that this parametrization validates the assumptions of Theorem 4.1. Let us choose $u^0, \ldots, u^{d_Z}$ in $supp(A)$, assumed to exist, such that these $d_Z + 1$ points are distinct and linearly independent. Define

$$U \in \mathbb{R}^{d_A \times d_Z}, \qquad U = \left(u^1 - u^0, \ldots, u^{d_Z} - u^0\right).$$

By construction, the columns of $U$ are linearly independent, so $U$ has full column rank, *i.e.*, $\text{rank}(U) = d_Z$.

Since $\Sigma$ is invertible, we have $\text{rank}(L) = \text{rank}(M_0 U)$. Moreover, $M_0$ is assumed to be full row rank of dimension $d_Z$. Therefore,

$$\text{rank}(M_0 U) = \min\{\text{rank}(M_0), \text{rank}(U)\} = \min\{d_Z, d_Z\} = d_Z.$$

Thus $M_0 U$ is square and invertible, which implies that $L$ is also invertible. This verifies the full-rank condition required in Theorem 4.1.

$\square$

# B EXPERIMENTS

## B.1 BASELINE METHODS

**Latent recovery** We perform evaluation of our latent recovery method against three existing methods: vanilla auxiliary constrastive learning (vACL) (Hyvärinen et al., 2019), iVAE (Khemakhem et al., 2020a) and first stage of Rep4Ex-CF (Saengkyongam et al., 2024). We use the same encoder and decoder architecture for each method, as well as the neural network architecture for each method to estimate the causal effects. Additionally, vACL includes a logistic regression head that we implement as an MLP with two hidden layers with ReLU activation, trained on cross-entropy loss. All three methods are implemented in our code that is appended to the submission. The network architecture for each method consists of the following blocks: 3 blocks of Linear - Batch normalization - LeakyRelu layers, with dropout at a rate of 0.2. The hidden dimensions are fixed at 16, 32 and 64 throughout both IMCA and extrapolation experiments.

**IV baseline comparison** We use the implementation of DeepGMM Bennett et al. (2019), KIV Singh et al. (2019) and DFIV Xu et al. (2021) provided in `https://github.com/liyuan9988/DeepFeatureIV`. We include an adapted version in our code, particularly new model specs as well as our data generative process. For the dSprites experiments we use an Image extractor (Table 3) for both DeepGMM and DFIV with a similar architecture as the encoder used for first stage of our method.

| ConvBlockDown($C_{in} \to C_{out}$) | Operations |
|---|---|
| Conv2d($C_{in} \to C_{out}$, kernel=3, stride=2, padding=1) | Downsampling conv |
| BatchNorm2d($C_{out}$) | Normalization |
| Activation (LeakyReLU(0.2) by default) | Non-linearity |
| Dropout2d(0.1) | Regularization |

Table 2: Definition of ConvBlockDown.

| Layer | Output Shape |
|---|---|
| Input ($1 \times 64 \times 64$) | $1 \times 64 \times 64$ |
| ConvBlockDown($1 \to 32$) | $32 \times 32 \times 32$ |
| ConvBlockDown($32 \to 64$) | $64 \times 16 \times 16$ |
| ConvBlockDown($64 \to 128$) | $128 \times 8 \times 8$ |
| ConvBlockDown($128 \to 256$) | $256 \times 4 \times 4$ |
| Flatten | 4096 |
| Dense($4096 \to 6$) | 6 |

Table 3: Image feature extractor used for DeepGMM, DFIV, and InfoIV in the dSprites experiment.

## B.2 IMCA DATA GENERATIVE PROCESS

**Injectivity of $g_0$.** Our identifiability result stated in Theorem 4.1 relies on the assumption that the ground-truth mixing function $g_0$ is injective. To enforce this property in our data-generating process, we use LeakyReLU activations and initialize the weight matrices of the linear layers to be full-rank. Particulary, $g_0$ has 2 hidden layers of dimension $[32, 64]$. Similarly, ground-truth causal effect $f_0$ is a 2 hidden layers neural network with tanh activations.

## B.3 DSPRITES DATA GENERATIVE PROCESS

We now describe the data generative process for dSprites data. We first sample a proxy between instrument and latent factors in order to avoid inverting the causal direction by defining the instrument as a direct function of the latent variable.

1. Sample a proxy variable $Q$ uniformly in a ball around the extremal values of $Z$.

2. Map $Q$ to the nearest existing latent value to define the latent features $Z$.

3. Compute the instrument $A$ as a nonlinear mapping of the components of $Q$ except for the one associated with position-y.

4. Obtain the observed treatment $X$ as the corresponding images from the dSprites dataset.

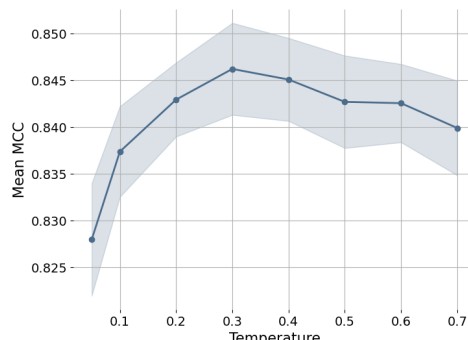

Figure 4: Average MCC on validation set (1,000 samples) over 20 runs for each temperature value. $d_A = 10, d_Z = 8, d_X = 12$. The training set includes 5,000 data points and the encoders are trained for 50 epochs on InfoNCE loss solely. Light blue area represents the 90% confidence interval.

5. Define the outcome as

$$Y = f_{\text{struct}}(Z) + \rho(posY - 0.5) + \eta,$$

where $f_{\text{struct}}$ is a randomly initialized neural network, $\rho$ is the confounding strength $\in [0, 1]$, and $\eta$ is Gaussian noise.

### B.4 INFOIV HYPERPARAMETERS TUNING

One advantage of our method over autoencoder-based approaches is that it depends on only a single hyperparameter: the temperature in the InfoNCE loss. We tune this parameter by evaluating the validation MCC 4, and notice the best performance is achieved with a temperature of 0.3, which we use for all subsequent experiments.

As mentioned earlier, we also explored adding a reconstruction term to our loss by training a decoder (mirrored architecture to the encoder) to reconstruct the input $X$. The resulting loss is:

$$\mathcal{L}(\phi, \psi, W) = \mathcal{L}_{\text{NCE}}(\phi, W) + \lambda_{\text{rec}} \|\psi \circ \phi(X) - X\|^2.$$

We conducted a study on the IMCA dataset, evaluating the learned latents against the ground truth using the MCC metric for different values of $\lambda_{\text{rec}}$. The latent features were then used in the second step of InfoIV-2SLS for causal effect estimation, which we evaluated using the out-of-sample MSE ($MSE_{oos}$, Figure 5). While values of $\lambda_{\text{rec}} > 1$ generally improve the consistency of the learned representation (increasing MCC by up to 0.2), they also lead to a deterioration in causal effect estimation, raising the MSE by an average of $1.5 \times 10^{-2}$.

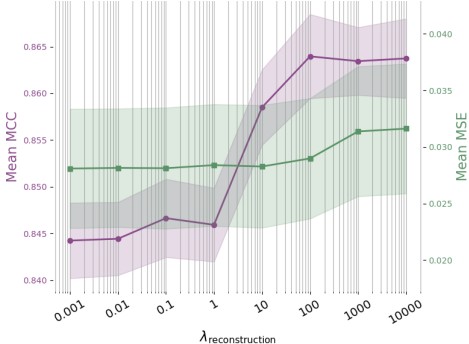

Figure 5: Average MCC (*in purple*) and out-of-sample MSE (*in green*) per reconstruction regularization parameter. The temperature for the InfoNCE loss term is fixed at 0.3.

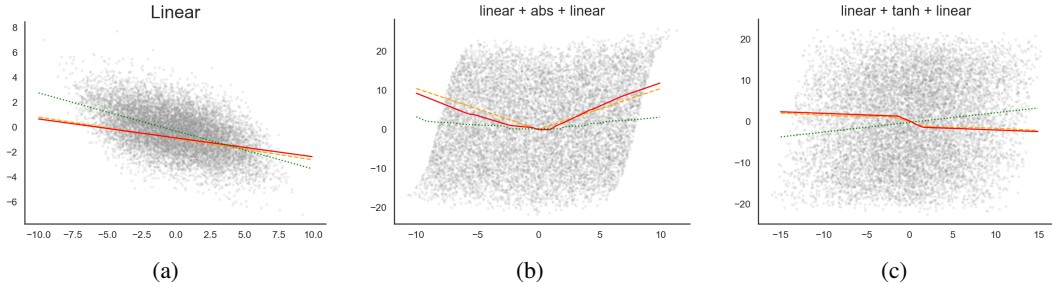

Figure 6: Estimated causal effect with InfoIV-CF (*in red*), ground-truth causal effect (*in orange*), ERM model (*in green*), (Z;Y) (*in grey*).

## B.5   EXTRAPOLATION PLOTS

We additionally evaluate our method in a setting where both $Z$ and $X$ are scalar, while $A$ is sampled from a two-dimensional uniform distribution. Figure 6 shows the learned causal-effect. We consider three scenarios: a) corresponds to the case of a linear causal effect; b) corresponds to a nonlinear causal effect implemented as a linear layer with hidden dimension 16, followed by an *abs* activation and a final linear layer; and c) corresponds to a similar architecture where the nonlinear activation is the $\tanh$ function instead of *abs*. We follow our standard training procedure for InfoIV-CF. InfoIV recovers the ground-truth causal effect $f_0$ up to an affine indeterminacy that arises from latent variable estimation. To account for this, we learn an affine transformation that aligns the estimated latent representation with the ground-truth $Z$, and we report the causal effect after applying this transformation. For comparison, we also fit an ERM model mapping the ground-truth $Z$ to the outcome $Y$. The ERM estimator fails to recover the causal effect, as $Z$ is confounded with the residual variation in $Y$. Importantly, despite the affine indeterminacy, our method still yields a valid estimate of the causal relationship from the observed $X$ to $Y$.

