# OpenReview forum: "Contrastive Learning Recovers Causal Features for Instrumental Variable Regression"
_ICLR.cc/2026/Conference — Submitted to ICLR 2026_

### Official Review · Reviewer_r75n · 2025-10-29

**Soundness:** 3
**Presentation:** 3
**Contribution:** 3
**Rating:** 6
**Confidence:** 4

**Summary:**

This paper addresses a significant and realistic extension of the classic IV setting by considering the case where the treatment variable is latent and only observed through a nonlinear, potentially high-dimensional transformation. The authors propose a novel contrastive learning framework that leverages insights from Independent Component Analysis (ICA) to recover the latent treatment variable up to an affine transformation. They theoretically prove that the recovered representation maintains compatibility with classical IV estimation techniques.
Comprehensive empirical results evaluates the robustness of the learned estimators under distribution shift conditions—an important practical consideration.

**Strengths:**

1. A key strength is the creative integration of ideas from Independent Component Analysis (ICA) into a contrastive learning framework for causal estimation. This cross-disciplinary approach is innovative and provides a principled way to address the identifiability challenge.

2. The paper provides a crucial theoretical proof that the representation recovered by their method remains compatible with classical IV estimators.

3. The empirical analysis is robust, demonstrating the framework's effectiveness with multiple classical estimators (Control Function and 2SLS). Importantly, the evaluation includes performance under distribution shifts, which is critical for assessing real-world robustness and is often overlooked.

**Weaknesses:**

1. The assumptions underlying this model setting appear somewhat restrictive, as they rest on the hypothesis that the observed variable X is solely influenced by the latent treatment. Given that the core assumptions of the instrumental variable approach are already challenging to verify, the justification for introducing additional assumptions concerning X becomes even more tenuous.

2. The logic of the paper is hindered by some typos, compromising the reader's ability to follow its arguments. The details about the typos can be found in questions.

**Questions:**

1. There are some inconsistent notations in paper writing. For example, in Line 168, the IV is notated as Z. However, the latent treatment is introduced after Line 178. I think in Line 168, it should be X.

2. In Line 176, it says "however, X has to be available at test time." Since the availability of X is a natural requirement of IV regression, I think it does not form as the shortcoming of control function method.

3. In Lemma 4.3, the conclusion may cause some confusion. The function $\hat{f}$ is learned on the representation $\tau(Z)$. What does it imply by combining the ground truth $f_0$ with $\tau(Z)$.

4. The figure of framework is rough and simple. I suggest the authors to refine it and improve the presentation.

---

> ### Author Response · Authors · 2025-11-19
> **Answer to reviewer r75n**
>
> We thank the reviewer for recognizing the novelty of integrating ICA with IV methods, the strength of our theoretical guarantees for compatibility with classical IV estimators, and the robustness of our empirical evaluation.
>
> > There are some inconsistent notations in paper writing. For example, in Line 168, the IV is notated as Z. However, the latent treatment is introduced after Line 178. I think in Line 168, it should be X.
>
> We acknowledge the confusion that might arise from the term "latent treatment", which we now replaced in the manuscript by "latent features" or "latent variables" when referring to $Z$, and we are now consistently refering to $X$ as treatment.
>
> > In Line 176, it says "however, X has to be available at test time." Since the availability of X is a natural requirement of IV regression, I think it does not form as the shortcoming of control function method.
>
> We thank the reviewer for catching this typo. The intended point is that a shortcoming of the control function approach is its requirement that the **instrument** is available at test time, whereas 2SLS does not have this requirement. The manuscript was modified accordingly.
>
> > The assumptions underlying this model setting appear somewhat restrictive, as they rest on the hypothesis that the observed variable X is solely influenced by the latent treatment. Given that the core assumptions of the instrumental variable approach are already challenging to verify, the justification for introducing additional assumptions concerning X becomes even more tenuous.
>
> Here, the main bottleneck, that is inherent to IV regression, is that the instrument carries sufficient information about $X$ (indirectly through $Z$). However, instead of directly using the instrument for $X$, we first learn a leatent representation that contains all relevant information of $X$ given the instrument $A$. In the worst-case, we cannot compress any information and $Z$ is as high-dimensional as $X$, corresponding to the assumptions in other frameworks (e.g., DFIV, DeepGMM).
>
> > In Lemma 4.3, the conclusion may cause some confusion. The function $\hat{f}$ is learned on the representation $\tau{Z}$. What does it imply by combining the ground truth $f_0$ with $\tau{Z}$.
>
> The reviewer notes an important point of the paper that we now clarified. Our goal is to learn the ground-truth causal effect of $X$ on $Y$ given by $f_0 \circ g_0^{-1}$. We added a paragraph in the text following Lemma 4.3 showing that $\tau$ is canceled out for estimating the causal effect of $X$ on $Y$ (lines 308 to 313), i.e., we can recover the causal effect of $X$ on $Y$ as follows:
> $$\hat{f} \circ \phi = \hat{f} \circ \tau \circ g_0^{-1} = f_0 \circ g_0^{-1}.$$
>
> > The figure of framework is rough and simple. I suggest the authors to refine it and improve the presentation.
>
> Thank you for this comment. We will revise and polish all figures for the next iteration of the paper.

---

### Official Review · Reviewer_kxh5 · 2025-10-30

**Soundness:** 3
**Presentation:** 2
**Contribution:** 3
**Rating:** 6
**Confidence:** 4

**Summary:**

This paper studies causal effect estimation in scenarios with unobserved confounders where the treatment variable is latent rather than directly observed. Inspired by Independently Modulated Component Analysis (IMCA), the authors propose InfoIV, a general contrastive learning framework designed to recover latent treatment variables for causal effect estimation.

**Strengths:**

1.The authors consider a more realistic setting where the treatment variable is unobserved, extending the applicability of instrumental variable regression.

2.The paper provides a coherent set of assumptions, definitions, and theoretical proofs to establish the validity of the InfoIV framework.

3.The method is conceptually simple and easy to understand.

**Weaknesses:**

1.It would be helpful to include a demonstrative example to illustrate why recovering a latent treatment variable Z is necessary when an observed variable X already exists. This would help clarify the motivation and conceptual necessity of introducing a latent treatment representation.

2.In Equation (6), the authors employ an encoder to recover Z from X. While this is an interesting design, it remains unclear why a contrastive learning paradigm can reliably recover the latent variable. From an empirical perspective, variational autoencoders (VAEs) or their variants are commonly used for this purpose.

3.As shown in Table 1, InfoIV does not perform strongly in causal effect estimation. A more detailed analysis is needed to understand the underlying causes. In particular, the authors should explore why InfoIV performs well on image data but poorly on tabular data.

4.In Section 4.3, InfoIV-CF assumes that the effect of the instrumental variable A on the treatment variable Z is linear, while in Section 5.1, the simulated data uses a nonlinear relationship between A and Z. This inconsistency should be clarified or justified.

**Questions:**

See Weakness

---

> ### Author Response · Authors · 2025-11-19
> **Answer to reviewer kxh5 (1/2)**
>
> We thank the reviewer for acknowledging the coherence of our theoretical proofs establishing the validity of our framework and his appreciation of a more realistic setting extending the applicability of instrumental variable regression.
>
> > It would be helpful to include a demonstrative example to illustrate why recovering a latent treatment variable Z is necessary when an observed variable X already exists. This would help clarify the motivation and conceptual necessity of introducing a latent treatment representation.
>
> We now include in the main text a real-world example motivating the assumption of latent structure on the treatment variable $X$:
> *"A practical example is a post-surgery CT scan, where the observed image $X$ reflects underlying factors $Z$ (e.g., surgical precision or tissue response), and the outcome corresponds to the patient’s recovery. Prior works in causal inference have also considered high-dimensional structured treatments, including text, images (Kaddour et al., 2021) or even graphs en graphs (Harada & Kashima, 2021)."*
> Further, the strength of the latent assumption shows in our experiment on the dSprites dataset (Figure 3) in which InfoIV has an advantage over all baselines.
>
> > In Equation (6), the authors employ an encoder to recover Z from X. While this is an interesting design, it remains unclear why a contrastive learning paradigm can reliably recover the latent variable. From an empirical perspective, variational autoencoders (VAEs) or their variants are commonly used for this purpose.
>
> Our experiments include a baseline comparison against a state-of-the-art identifiable VAE-based method, which our approach outperforms while using substantially fewer parameters (since we do not have to train a decoder). VAE-based methods are known to offer stronger identifiability guarantees ([1], Theorem 2), enabling recovery of the ground-truth latent structure up to permutation and pointwise transformations. In contrast, our contrastive approach recovers latents only up to affine and pointwise transformations (Theorem 4.1). Nevertheless, Lemma 4.3 establishes that this level of indeterminacy is sufficient for identifying the causal effect from $X$ to $Y$. Further, since VAE-based approaches have to fully reconstruct $X$, we conjecture that they keep more information in the latent representation that is irrelevant for the IV task, which would explain the decrease in causal effect estimation performance when adding a decoder (Appendix B.4).
>
> More broadly, our work aims to leverage (and to some extend explain) the recent progress in self-supervised representation learning, where contrastive objectives are used to learn robust latent representations that transfer effectively to downstream tasks, e.g., in large-scale models such as CLIP [2].
>
>
> - [1] I. Khemakhem et al. Variational autoencoders and nonlinear ica: A unifying framework. AISTATS 2021
> - [2] A. Radford et al. Learning Transferable Visual Models From Natural Language Supervision.

---

> ### Author Response · Authors · 2025-11-19
> **Answer to reviewer kxh5 (2/2)**
>
> > As shown in Table 1, InfoIV does not perform strongly in causal effect estimation. A more detailed analysis is needed to understand the underlying causes. In particular, the authors should explore why InfoIV performs well on image data but poorly on tabular data.
>
> To showcase that our method has a comparable performance to the baselines, we added a new result to Table 1, which shows the result from simple 2SLS with neural network estimators. We note that the error of this baseline is one order of magnitude worse than for InfoIV (~$10^{-2}$) while InfoIV and the remaining baselines have errors ~$10^{-3}$ with the exception of DFIV that performs slightly better for $\rho = 0.1$. Please note that DFIV has a more refined optimization step (by jointly training the neural networks of both stages, enabling flexible nonlinear first- and second-stage mappings) which is why we suspect that it performs better on this task. When we switch to the image dataset, the strength of our method is more pronounced since we can strongly reduce the intrinsic dimensionality with the learned encoder, and the mixing process is substantially more complex. Here, reconstruction-based methods need to keep all information about $X$, which--indicated by our results--harms the IV regression task, as shown in our ablation (Appendix B.4), in which we experimented with adding a reconstruction term and a decode to our approach.
>
> > In Section 4.3, InfoIV-CF assumes that the effect of the instrumental variable A on the treatment variable Z is linear, while in Section 5.1, the simulated data uses a nonlinear relationship between A and Z. This inconsistency should be clarified or justified.
>
> We thank the reviewer for pointing out this lack of clarity in the presentation. The data-generating process for the extrapolation experiment—where InfoIV-CF is used—is detailed in Section 5.3 and specifies that the relationship between $A$ and $Z$ is linear via a sampled full-rank matrix. We have revised the text to ensure each experiment clearly references its corresponding setup.

---

### Official Review · Reviewer_qKPF · 2025-10-30

**Soundness:** 2
**Presentation:** 3
**Contribution:** 2
**Rating:** 2
**Confidence:** 5

**Summary:**

The paper considers an instrumental variable (IV) regression setting in which the treatment is hidden and only a non-linear transform of it is observed. The paper then proposes to apply a two-step approach called InfoIV: (1) Use a causal representation learning approach to estimate a non-linear transform of the hidden treatments and (2) use the resulting estimated hidden treatments in a two-stage least squares approach.
The authors additionally combine the first step (learning the hidden treatments) with a control function approach that allows them to perform extrapolation on the instrument domain.
Finally, InfoIV is evaluated in three experimental settings: IV on tabular data, IV on image data and extrapolation on the domain of the instrument.

**Strengths:**

- Good high-level presentation: It was easy to read and understand the paper; it is also nice that the authors provide detailed background on IV for readers less familiar with it.
- Interesting application of causal representation learning: Overall, the idea of using causal representation learning for a concrete problem like this is interesting.

**Weaknesses:**

- Unclear objective and motivation: While the authors are clear about the setting they consider, my main concern with this work is that the actual problem the authors want to solve is not clear. Given that the variable Z is unobserved, it is unclear to me why we would want to estimate f_0 in this setting. Additionally, given that we actually cannot identify Z but only a coordinate-wise non-linear transformation, it is even less clear why identifying the function in Eq. (7) is useful (this also shows in the experiments, I believe; see below). The extrapolation problem considered in the second half seems much more clearly defined.
- Concerns about experimental evaluation:
  - Recovering latent representation experiments: Why does it make sense to use mean correlation coefficients here? Since the hidden components are only identifiable up to permutations and non-linearities, it seems that one would need a very different evaluation metric.
  - IV experiments: In these experiments, the authors consider an evaluation metric where f_0 is a function of X and not of the hidden variable Z. All baselines should therefore be misspecified in this model, and even the proposed InfoIV method is not intended for this use case. At the very least, regular least-squares regression of Y on X should be applied here (which should actually perform best on the loss if I understand it correctly).
  - Extrapolation: For the extrapolation experiment, I do not understand why in Fig. 2(b) the representations of the MMR (Rep4Ex) approach are worse but the extrapolation of Rep4Ex performs better (aren't the methods InfoIV and Rep4Ex the same apart from how they recover the latent representation, so better representations should lead to better extrapolation?).
- Better clarity on assumptions and conditions would be helpful:
  - Independence between A and eps is often written as required for IV although it is too strong; instead only E[eps|A] = 0 is needed (same is true for least-squares regression with X instead of A). Please be precise about which of them you need.
  - Direct effect from A to X is not needed for IV but is needed for the extrapolation (I think), so it should not appear in Assumption 2.1 (it is also a much stronger assumption than P(X|A) not constant).
  - In Section 4.3, I found it hard to follow which additional assumptions are needed (and which assumptions might not be needed) in the comparison with Rep4Ex.

**Questions:**

- What application do you have in mind when you want to estimate the causal effects of the latent variables on the outcome?

---

> ### Author Response · Authors · 2025-11-19
> **Answer to reviewer qKPF (1/2)**
>
> We thank the reviewer for their encouraging comment that we are considering an interesting application for causal representation learning. Below we would like to answer to the reviewers questions and concerns.
>
> > **Unclear objective and motivation**
> > [...] Given that the variable Z is unobserved, it is unclear to me why we would want to estimate f_0 in this setting.
>
> In order to avoid confusion about our setting, we now refrain from using the term "latent treatment" for $Z$ since, as the reviewer points out correctly, we can only determine $f_0$ up to some indeterminacy. Our main goal is to learn the treatment effect of the high-dimensional treatment $X$ on $Y$, similar to, e.g., DFIV or DeepGMM. However, we assume that there exists a more concise latent representation of $X$ that contains the relevant information that $X$ has about $Y$ as common in causal representation learning. Our experiments on the dSprites dataset confirm that this assumption is sensible and leads to improved results.
>
> > [...] even less clear why identifying the function in Eq. (7) is useful (this also shows in the experiments, I believe; see below).
>
> Thank you for this comment. We added a derivation (lines 308-313) after Lemma 4.3 that explicitly links our representation learning result in Theorem 4.1 and the IV result in Lemma 4.3 (Eq. (7)), i.e., we can recover the causal effect of $X$ on $Y$ as follows:
> $$\hat{f} \circ \phi = \hat{f} \circ \tau \circ g_0^{-1} = f_0 \circ g_0^{-1},$$
>
> where we have that $f_0(g_0^{-1}(x)) = f_0(z)$ according to the assumed generative mechanism. Note that the indeterminacy from the latent recovery cancels with the learned function $\hat{f}$.
>
> > **Concerns about experimental evaluation**
> > Why does it make sense to use mean correlation coefficients here?
>
> The MCC is computed between the ground-truth and estimated latent representations after aligning them via the optimal affine transformation. This is now clearly mentioned in the text (line 410). This makes the metric invariant to linear and pointwise transformations—exactly matching the identifiability level of our method. For completeness, we point the reviewer to metrics/mcc.py which was obtained from the implementation of [1].
>
> > IV experiments: In these experiments, the authors consider an evaluation metric where f_0 is a function of X and not of the hidden variable Z. [...] regular least-squares regression of Y on X should be applied here (which should actually perform best on the loss if I understand it correctly).
>
> Thank you for pointing out the typo: in our setting the structural function depends on the latent structure $Z$, and the correct expression for our *out-of-distribution* evaluation metric is therefore (replacing $x_i$ with $z_i$)
>
> $$MSE_{\mathrm{oos}} = \frac{1}{n} \sum_{i=1}^{n} \|\hat{y}_i - f_0(z_i)\|^2 \;.$$
>
> In other words, we evaluate the predictor by comparing the estimated outcome with the ground-truth *unconfounded* outcome, the same evaluation metric is widely used in IV literature, e.g., in DeepGMM or DFIV to which we compare. We hope that after correcting this typo it is clear why ERM estimator should not be expected to outperform the other baselines that we consider.
>
> > Extrapolation: [...] why in Fig. 2(b) the representations of the MMR (Rep4Ex) approach are worse but the extrapolation of Rep4Ex performs better
>
> Please note that we use InfoIV-CF for the results in Fig. 4(b), which achieves reconstruction performance comparable to Rep4Ex with better consistency. Rep4Ex, however, enforces independence between the learned latent and the confounder by explicitly leveraging the instrument, which we suspect contributes to its advantage in the extrapolation task. The strength of our approach lies in demonstrating that contrastive learning can be effectively applied to both nonlinear IV estimation and extrapolation—two tasks that are typically addressed using VAE-based frameworks. In our ablation in Appendix B.4, we experiment with adding a reconstruction term and a decoder (leading to the improved reconstruction in Figure 4(b)), but note that this harms the causal inference task. We suspect that this may be due to the fact that we need to retain more information about $X$ that is irrelevant in the context of estimating the treatment effect, due to the reconstruction term.
>
> - [1] I. Khemakhem et al., *ICE-BeeM: Identifiable Conditional Energy-Based Deep Models Based on Nonlinear ICA*. NeurIPS 2020.

---

> ### Author Response · Authors · 2025-11-19
> **Answer to reviewer qKPF (2/2)**
>
> > **Clarifications on assumptions**
> > Independence between A and $\varepsilon$ [...]
>
> Since our proof of causal effect recoverability (Lemma 4.3) builds on the framework of Newey et al. 2003, we adopt the same assumption on the instrument, stated as Assumption 2.1: $\mathbb{E}[ \varepsilon∣A ] = 0$.
> This condition is sufficient to obtain the derivation in Theorem 2.2, which in turn motivates the two-stage approach to IV regression. Since this does not imply that $\varepsilon$ and $A$ are independent, we removed the *unconfoundedness* claim in our paper.
>
>
> > Direct effect from A to X is not needed for IV but is needed for the extrapolation (I think), so it should not appear in Assumption 2.1 (it is also a much stronger assumption than P(X|A) not constant).
>
> The assumption $P(X|A)$ not constant is indeed sufficient for IV regression.
>
>
> > In Section 4.3, I found it hard to follow which additional assumptions are needed (and which assumptions might not be needed) in the comparison with Rep4Ex.
>
> In comparison to the IV-setting, we require that the relationship from $A$ to $Z$ is linear and the corresponding mixing matrix is full rank. This assumptions is necessary for extrapolation ([2], Proposition 1). We rely on the same assumptions as Rep4Ex. In particular, Corollary 4.4 shows that these assumptions also hold for our contrastive objective.
>
> > Q: What application do you have in mind when you want to estimate the causal effects of the latent variables on the outcome?
>
> Our goal is to compute the causal effect of high-dimensional treatment $X$ on $Y$ while assuming latent structure of $X$. We included a real-world example in the main text as well as a paragraph putting our framework in context with other related works on high-dimensional treatment (lines 47 to 53):
> *"A practical example is a post-surgery CT scan, where the observed image $X$ reflects underlying factors $Z$ (e.g., surgical precision or tissue response), and the outcome corresponds to the patient’s recovery. Prior works in causal inference have also considered high-dimensional structured treatments, including text, images (Kaddour et al., 2021) or even graphs en graphs (Harada & Kashima, 2021)."*
>
> As detailed in the response to the general setting, we now included a new remark linking our theoretical results of latent recovery and estimation of the treatment effect after Lemma 4.3.
>
> - [2] S. Saengkyongam et al., *Identifying Representations for Intervention  Extrapolation*. ICLR 2024.

---

### Official Review · Reviewer_AcMu · 2025-10-31

**Soundness:** 2
**Presentation:** 2
**Contribution:** 1
**Rating:** 2
**Confidence:** 4

**Summary:**

This paper explores the application of ICA-based identifiable deep generative models, particularly Ice-beem (Khemakhem 2022b), to instrumental variable (IV) settings in causal inference. It proposes a framework for identifying the treatment and estimating causal effects when treatments may be latent variables, drawing analogies to traditional IV regression through a two-stage procedure. The approach assumes noiseless injection from low-dimensional treatments to higher-dimensional covariates and demonstrates results in simulated experiments, including image-based scenarios.

**Strengths:**

The paper is relatively well-presented and easy to read, with clear figures and logical flow in sections.

Applying ICA-based identifiable deep generative models to instrumental variable (IV) settings represents a meaningful research direction.

**Weaknesses:**

- **Mis-conceptualized Problem Setup**: The core issue is a fundamental misunderstanding of causal interventions in IV frameworks. Treatments cannot be hidden; interventions require *full observability and control* over the variable in question. For instance, randomized controlled trials (RCTs) demand extensive effort to *design treatment assignment mechanisms*, which drives their high cost. While theoretically possible, the paper's latent treatment assumption lacks grounding in real-world applications. The authors should identify or propose a plausible scenario where treatments are truly unobservable—none is evident here.

   - **Inconsistent Notation**: Symbol usage is erratic and confusing, likely signaling the conceptual issue above. In the Introduction and Figure 1, $Z$ denotes the treatment; in Section 2, it shifts to $X$; and in Section 3, it reverts to $Z$. This inconsistency is extremely confusing.
   - **Experiment (Section 5.2)**: The experiment treats observed images $X$—not latent variables—as the intervention target, directly contradicting the paper's latent treatment premise. This is unsurprising given the earlier critique: hidden treatments are incompatible with standard intervention paradigms.

- **Weak Connection to IV Regression**: Identification and estimation in IV rely on observed treatments. In theory, it is okay to establish that treatments are recovered from ICA, but then, the method diverges fundamentally from IV. The paper's claim of a "connection" via latent recovery is superficial, resembling vague analogies (e.g., a two-stage least squares procedure).

- **Unreasonable Noiseless Injection Assumption**: The model assumes deterministic injection from a low-dimensional treatment to high-dimensional covariates $X$, which is mathematically permissible but practically implausible. Even if their dimensionality is the same, in real settings, treatments are rarely deterministically related to covariates. Also, consider causal directions: under what conditions does a treatment affect *all* covariates, deterministically or up to some noise? None comes to mind. Introducing additive noise (a more realistic assumption) would likely invalidate the current estimation approach.

- **Omission of Key Related Work**: The paper neglects recent advances in ICA-based causal effect methods, based exactly on Independently Modulated Component Analysis (IMCA; Khemakhem et al., 2020a,b). Notable omissions include:
  - Wu and Fukumizu (2022): "Intact-VAE: Identifying and Estimating Causal Effects under Limited Overlap" (ICLR 2022). This is the *first* work applying ICA-based identifiable deep models to causal inference.
  - Xu et al. (2024): "Causal Inference with Conditional Front-Door Adjustment and Identifiable Variational Autoencoder" (ICLR 2024).

**Questions:**

Please refer to the points in Weaknesses.

---

> ### Author Response · Authors · 2025-11-19
> **Answer to reviewer AcMu (1/2)**
>
> We thank the reviewer for their critical evaluation of our paper, which helped us to improve our manuscript in the following aspects.
>
>
> >  This paper explores the application of ICA-based identifiable deep generative models
>
> We would like to clarify that we **do not use a deep generative model** as stated explicitely: "*To avoid the problem of having to learn a powerful conditional generative model, we instead propose to approximate the conditional distribution in the latent space.*".
> One of the key ideas of our framework is to use contrastive learning (see Section 4.1). As an ablation, we considered adding a decoder in the latent recovery phase which, however, decreases the performance of our method on the causal effect estimation task (see Appendix B.4).
>
>
> > **Mis-conceptualized Problem Setup**
> > Treatments cannot be hidden
>
> We acknowledge the confusion that might arise from the term "latent treatment", which we now replaced in the manuscript by "latent features" or "latent variables" when referring to $Z$. In particular, we do not assume that the "treatment is hidden", but instead that the treatment $X$ is high-dimensional and assumes a latent structure that interacts with the outcome variable. Such a latent variable assumption is standard in representation learning, especially in causal representation learning and ICA. Our experiments on the dSprites dataset confirm that this assumption is sensible and leads to improved results. Moreover, the assumption of having a high-dimensional (image) treatment variable is in line with other recent works, such as DFIV or DeepGMM (which do not assume the existence of a latent structure), or the following works that assume structured treatment variables [1] or graphs [2], which we now cite in our paper (cf. line 53).
>
> > The authors should identify or propose a plausible scenario where treatments are truly unobservable—none is evident here.
>
> Besides renaming $Z$ throughout our paper, we now added the following paragraph to the introduction to clarify our setting:
> "*We focus on a setting where the treatment $X$ is high-dimensional but driven by a lower-dimensional latent structure $Z$ through an injective mixing function $g_0{:} \; \mathcal{Z} \rightarrow \mathcal{X}$, i.e $X := g_0(Z)$. The outcome variable $Y$ depends on these latent factors rather than on the raw high-dimensional observations ($X$ influences outcome only through latents). A practical example is a post-surgery CT scan, where the observed image $X$ reflects underlying factors $Z$ (e.g., surgical precision or tissue response), and the outcome corresponds to the patient’s recovery.*"
>
> Further, we added a remark after Lemma 4.3 (lines 308 to 313) that explicitly links our representation learning result in Theorem 4.1 and the IV result in Lemma 4.3, i.e., we can recover the causal effect of $X$ on $Y$ as follows:
> $$\hat{f} \circ \phi = \hat{f} \circ \tau \circ g_0^{-1} = f_0 \circ g_0^{-1},$$
> where we have that $f_0(g_0^{-1}(x)) = f_0(z)$ is the true **unconfounded** outcome, according to the assumed generative mechanism. Note that the indeterminancy from the latent recovery cancels with the learned function $\hat{f}$.
>
> - [1] Kaddour et al. *Causal Effect Inference for Structured Treatments*. NeurIPS, 2021
> - [2] Harada and Kashima. *Graphite: Estimating individual effects of graph-structured treatments*. CIKM, 2021

---

> ### Author Response · Authors · 2025-11-19
> **Answer to reviewer AcMu (2/2)**
>
> > Inconsistent Notation
>
> We fixed our notation such that we always refer to $X$ as the treatment.
>
> > Experiment (Section 5.2): The experiment treats observed images —not latent variables—as the intervention target
>
> We would like to correct this misunderstanding. We do intervene on the latent variables (rotation, color, shape, etc.), as stated in Appendix B.3. This intervention is reflected one-to-one in the corresponding image ($X$) that is provided to the methods, which, however, cannot access the latent features directly. This experimental setup differs from DFIV where the authors directly generate $Y$ based on the pixel values and not the latent variable.
>
> > Weak Connection to IV Regression: Identification and estimation in IV rely on observed treatments. In theory, it is okay to establish that treatments are recovered from ICA, but then, the method diverges fundamentally from IV. The paper's claim of a "connection" via latent recovery is superficial, resembling vague analogies (e.g., a two-stage least squares procedure).
>
> As outlined above, we consider the case of nonlinear IV, similar to DFIV, where we can have a treatment $X$ that is high-dimensional (e.g., an image). We show in Theorem 4.1 and Lemma 4.3 that we can recover the true function mapping $X$ to $Y$ by first recovering the latent features $Z$ (which we assume exist) and **subsequently applying classical nonlinear IV methods such as 2SLS** on the recovered latent $Z$. As stated above, the key idea is that the indeterminacy from the latent recovery cancels with the function $\hat{f}$ learned, e.g., via 2SLS.
>
> > Unreasonable Noiseless Injection Assumption: The model assumes deterministic injection from a low-dimensional treatment to high-dimensional covariates , which is mathematically permissible but practically implausible. [...]
>
> Thank you for this comment. It would be interesting to consider an extension of our work that weaken this assumption following the work of Khemakhem et al. 2020 (iVAE). However, we cannot straightforwardly build on this result as it requires a decoder and our work focuses on contrastive learning. We added a discussion of this aspect in the conclusion section.
>
>
> > Omission of Key Related Work: The paper neglects recent advances in ICA-based causal effect methods, based exactly on Independently Modulated Component Analysis (IMCA; Khemakhem et al., 2020a,b). Notable omissions include:
>
> Thank you for pointing out these works which leverage and extend the **iVAE** framework to the potential outcomes framework and in the context of the front-door adjustment. We added a discussion of these works in the introduction, however, we would like to emphasize that we (a) do not use a VAE but focus on **contrastive learning** which does not require a decoder (we compare to iVAE in Fig. 3b) and (b) study the setting of instrument variable regression and its connection to extrapolation.

---

### Author Response · Authors · 2025-11-19
**General Response to Reviewers and Updates to the Manuscript**

We would like to thank the reviewers for their constructive comments as well as their appreciation of our work. Reviewers praised the paper for its accessible presentation (AcMu, qKPF, kxh5), its "creative" and "interesting" integration of ICA/CRL ideas into an IV framework (r75n, qKPF), its solid theoretical guarantees (kxh5, r75n), and its robust empirical evaluation (r75n).

Here we would like to comment on the modifications that we made in the paper based on the reviewers feedback:
 - **Clarity on the goal of learning causal effects on latent variables and the notion of "latent treatment" (AcMu, qKPF, kxh5, r75n)**
 -> We added a paragraph clarifying the role of $Z$ as a latent representation (refraining from the term "latent treatment") of the observed treatment $X$ (lines 47-50) and additional derivation connecting Theorem 4.1 and Lemma 4.3 proving that the causal effect of $X$ on $Y$ can be recovered despite the inherent indeterminacy of the representation learning phase (lines 308-313), i.e.,
 $$\hat{f} \circ \phi = \hat{f} \circ \tau \circ g_0^{-1} = f_0 \circ g_0^{-1}.$$
 - **Results on low-dimensional data (Table 1) (kxh5)**
 -> We added a new baseline to Table 1, performing plain 2SLS on observed treatment $X$ based on neural networks with similar architecture as InfoIV-2SLS that we outperform by one order of magnitude---showcasing the our performance is comparable with the remaining baselines.
 - **Lack of a practical example illustrating our setting (AcMu, qKPF, kxh5)**
 -> We added a motivating real-world example where the treatment may correspond to a post-surgery CT scan, where the observed image $X$ reflects underlying factors $Z$ (e.g., surgical precision or tissue response, etc.). We further referenced prior work that considers high-dimensional structured treatments, providing context and motivation for our approach (lines 50-53).

We further polished the paper and added a discussion on the limitations of approach in the conclusion, corrected typos spotted by the reviewers, and improved the figures. More details to the reviewers questions are provieded in the individual responses.

---

> ### Author Response · Authors · 2025-11-27
>
> Dear Reviewers,
>
> Thank you again for your constructive remarks to our paper. We have carefully considered your feedback and updated our manuscript accordingly. In particular, (besides other aspects) we now align our motivation and setup more closely with other deep IV frameworks to address one of the main concerns that has been raised. Please let us know if our revisions and explanations resolve your concerns, as this will help us further improve our manuscript.

---

### Meta-Review · Area_Chair_krcZ · 2026-01-05

**Summary:**

The paper consider instrumental variable (IV) regression setting with treatment which is not directly observed but only a non-linear transformation of it is observed. A two-step method, InfoIV, has been proposed to identify the hidden treatment and estimate causal effects. In the first step, the method leverages ICA-based identifiable models based on Ice-beem to learn a causal representation corresponding to treating the treatment as a latent variable, drawing a clear analogy to classical IV regression. The framework assumes a noiseless mapping from low-dimensional treatments to higher-dimensional covariates and is evaluated through simulated experiments across multiple settings, including tabular data, and image-based problems.

**Reviewer Concerns:**

The reviewer concerns are partially addressed. The ones relating to latent confounder, and treatment may still remain to be . The authors engaged in discussion to change it to latent features. It still pose a gap between the latent variable and non-linear transformation of treatment.

Experiment evaluation is of another concern including the data generation procedure and interpretation of correlation coefficient.

The assumption of independence/conditional independence structure and direct effect outsider IV procedure raised by reviewer qKPF may still remain unclear.

The connections between IV regression in the discussion are still not too clear. The use and reference of deep generative modelling, as well as its adaptation, are also ambiguous, giving rise to the miscommunication.

**Reviewer Scores:**

Given the concerns of reviewers may not be fully addressed, the score may potentially not change.

---

### Decision · Program_Chairs · 2026-01-26

Reject